# Spider-silk-inspired strong and tough hydrogel fibers with anti-freezing and water retention properties

Shaoji Wu [1], Zhao Liu[1], Caihong Gong[1], Wanjiang Li[1], Sijia Xu[1], Rui Wen[1], Wen Feng [2] ✉, Zhiming Qiu[1] & Yurong Yan [1,3] ✉

Ideal hydrogel fibers with high toughness and environmental tolerance are indispensable for their long-term application in flexible electronics as actuating and sensing elements. However, current hydrogel fibers exhibit poor mechanical properties and environmental instability due to their intrinsically weak molecular (chain) interactions. Inspired by the multilevel adjustment of spider silk network structure by ions, bionic hydrogel fibers with elaborated ionic crosslinking and crystalline domains are constructed. Bionic hydrogel fibers show a toughness of $162.25 \pm 21.99$ megajoules per cubic meter, comparable to that of spider silks. The demonstrated bionic structural engineering strategy can be generalized to other polymers and inorganic salts for fabricating hydrogel fibers with broadly tunable mechanical properties. In addition, the introduction of inorganic salt/glycerol/water ternary solvent during constructing bionic structures endows hydrogel fibers with anti-freezing, water retention, and self-regeneration properties. This work provides ideas to fabricate hydrogel fibers with high mechanical properties and stability for flexible electronics.

Hydrogel fibers are of interest in flexible electronics for their stretchability, ionic conductive pathway, and the ability to construct three-dimensional structures from the bottom up[1–6]. However, the high water content of hydrogel-based materials weakens their intra- and intermolecular chain interactions, leading to poor mechanical properties of hydrogel-based materials[7–10]. Although toughening strategies such as directed freeze-casting, double network structure design, and multiple hydrogen bonding synergy have been developed for improving the mechanical property of hydrogel bulk, hydrogel fibers are difficult to apply these strategies due to limitations of spinning processes[11–14]. In addition, the high free water percentage in hydrogel-based materials causes them to exhibit environmental instability, losing their flexibility and conductivity at low-temperature and dry environments, especially for hydrogel fibers with ultra-high specific surface areas[15–22]. Therefore, it remains

a challenge to achieve both high mechanical properties and environmental tolerance of hydrogel fibers.

Spider silk is a natural fiber with ultimate toughness and excellent environmental tolerance, and biomimicking its relevant structures is helpful to fabricate desirable hydrogel fibers. For the high toughness of spider silk, theoretical and molecular dynamics simulations suggest that β-nanocrystalline domains with dense hydrogen bonds are the key[23,24]. Accordingly, Wu et al. prepared hydrogel fibers with high mechanical properties (tensile stress/strain of 11.76 MPa/210.20%) and super-shrinkage properties by biomimicking the soft-hard hydrogen bonding structure of spider silk[25]. Inspired by β-nanocrystalline domain crosslinked amorphous peptides, Liu et al. designed high-strength hydrogel fibers by doping zinc ions as additional crosslinking site into polyacrylic acid systems crosslinked with vinyl-functionalized silica nanoparticles[26]. Although the high crosslink density provided the

[1]School of Materials Science and Engineering, South China University of Technology, Guangzhou 510641, PR China. [2]Guangdong Medical Products Administration Key Laboratory for Quality Research and Evaluation of Medical Textile Products, Guangzhou 511447, PR China. [3]Key Lab of Guangdong High Property & Functional Polymer Materials, Guangzhou 510640, PR China. ✉e-mail: fengw@gttc.net.cn; yryan@scut.edu.cn

hydrogel fiber with high strength (261.00 MPa), the reported strain was undesirable as 49.20%. The current research on spider-silk-inspired robust hydrogel fibers is mainly surrounding on β-nanocrystalline domains[27,28]. In fact, various ions play an important role in the spinning process of spider silk, and they can adjust the mechanical properties of spider silk through ionic coordination and Hofmeister effects[29–33]. Moreover, the hydration effect of ions and the strong hydrogen bonding effect of proteins with rich polar groups contribute to the tolerance of spider silk to low temperatures and different humidity environments[34,35]. Apparently, the salt-regulated structural-performance paradigm of spider silk provides a great opportunity to develop strong and tough hydrogel fibers with environmental tolerance.

Inspired by the salt-regulated structural-performance paradigm from spider silks, a bionic structure engineering (BSE) strategy was presented to construct bionic hydrogel fibers with high mechanical properties and environmental tolerance. As a demonstration, anionic and crystalline domain crosslinked hydrogel fiber was constructed by utilizing the coordination of zirconium ions ($Zr^{4+}$) with polyacrylic acid (PAA) and the Hoffmeister effect-sensitive property of polyvinyl alcohol (PVA). Meanwhile, biomimicking the environmental tolerance mechanism of spider silk, hydroxyl-rich glycerol (Gly) was introduced, the strong hydrogen bonding effect of Gly could synergize with the hydration effect of ions for enhancing the environmental tolerance of the hydrogel fiber. The prepared bionic hydrogel fiber exhibited high toughness ($162.25 \pm 21.99$ MJ/m³) comparable to spider silk, and remained excellent mechanical

properties (stress > 50 MPa, strain > 200%) at −40 °C or 30%RH. Notably, just as ions were important in spider silk, adjusting the inorganic salt concentration could significantly change the modulus of the hydrogel fibers from gel to plastic levels ($3.74 \pm 0.16$ MPa to $118.53 \pm 5.49$ MPa). Moreover, the BSE strategy was universal and directly applicable to different combinations of Hoffmeister effect-sensitive polymers and inorganic salts, offering a solution for fabricating hydrogel fibers with high mechanical properties and environmental tolerance.

## Results

### Fabrication of hydrogel fibers based on the BSE strategy

Ions can contribute to the fabrication of natural spider silks with high mechanical properties through ionic coordination and Hoffmeister effects (Fig. 1a). Therefore, a BSE strategy was proposed to adjust the non-covalent interactions of polymers through inorganic salts. To successfully implement the corresponding bionic design, we employed an improved self-lubricating spinning strategy, which allowed us to freely design the hydrogel fiber network structure for the continuous fabrication of bionic hydrogel fibers[36]. For demonstration, the Hoffmeister effect-sensitive PVA and ion-coordinatable PAA were chosen as a model system. The kosmotropic sodium sulfate ($Na_2SO_4$) and high coordination number of $Zr^{4+}$ were used to construct crystalline domains and ionic crosslinks, respectively. In addition, Gly with strong hydrogen bonding effect was introduced to enhance the environmental tolerance of bionic hydrogel fibers. The preparation of

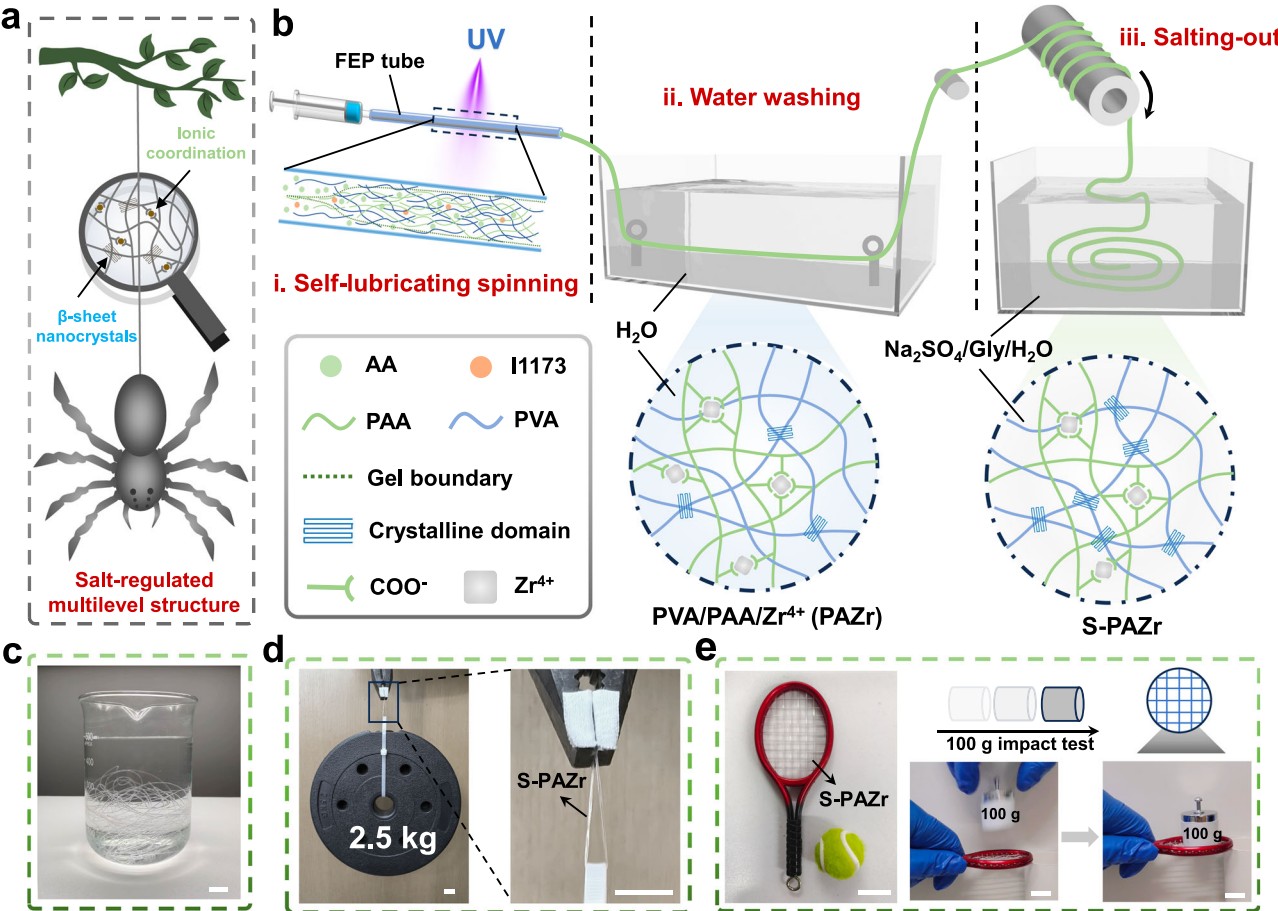

**Fig. 1 | Inspired by spider silks, a hydrogel fiber with ionic and crystalline domain crosslinking was designed by utilizing the ionic coordination and Hoffmeister effects of inorganic salts. a** Multilevel crosslinking structure of natural spider silk. **b** Schematic preparation of the S-PAZr hydrogel fiber.

**c** Optical images of collected S-PAZr hydrogel fibers (scale bar = 1.5 cm). **d, e** The S-PAZr hydrogel fiber lifted 2.5 kg weight and withstood the impact of a 100 g weight (scale bar = 1.5 cm).

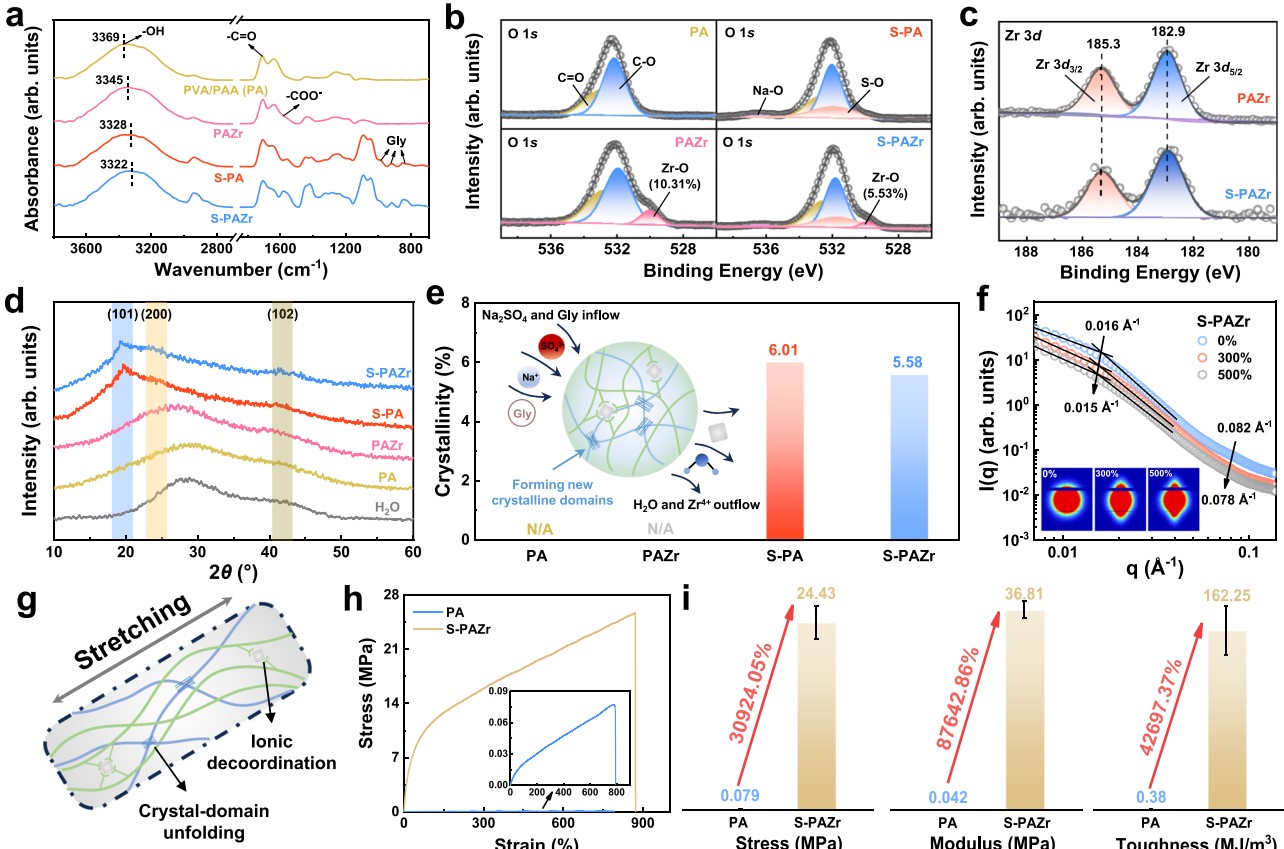

**Fig. 2 | Structural evolution and toughening mechanism of the S-PAZr. a** FTIR spectra of the PA, PAZr, S-PA, and S-PAZr. **b** O 1s XPS spectra of the PA, PAZr, S-PA, and S-PAZr. **c** Zr 3d XPS spectra of the PAZr and S-PAZr. **d** XRD profiles of the PA, PAZr, S-PA, and S-PAZr. **e** The crystallinity of the PA, PAZr, S-PA, and S-PAZr, and structural evolution of the S-PAZr during salting-out treatment. **f** SASX profiles of the S-PAZr at 0%, 300%, and 500% strain. **g** Synergistic energy dissipation by ionic crosslinking and crystalline domains during stretching. **h**, **i** A comparison for mechanical properties of PA and S-PAZr hydrogel fibers. Data were presented as mean ± SD ($n$ = 3 independent samples).

bionic hydrogel fibers was illustrated in Fig. 1b. First, the ionic cross-linked PVA/PAA/Zr$^{4+}$ (PAZr) hydrogel fiber was spun by an improved self-lubricating spinning strategy. Further, the PAZr hydrogel fiber was washed in a water bath to remove residual monomers followed by soaking in a Na$_2$SO$_4$/Gly/H$_2$O ternary solvent for solvent exchange. Solvent exchange in the Na$_2$SO$_4$/Gly/H$_2$O ternary solvent promoted the formation of PVA crystalline domains and removed some of the unstable ionic coordination, the PAZr hydrogel fiber was converted into the S-PAZr hydrogel fiber. As a result, the S-PAZr hydrogel fiber prepared by the BSE strategy exhibited excellent mechanical properties. The transparent S-PAZr hydrogel fiber could lift ~50,000 times its own weight (2.5 kg) after folded (Fig. 1c, d). Moreover, the hydrogel fiber allowed to be woven into a racket to withstand the impact of a 100 g weight falling freely from 5 cm height (Fig. 1e).

## Structural evolution and toughening mechanism of the S-PAZr hydrogel fiber

Ionic coordination and crystalline domain crosslinking are intertwined in the S-PAZr. To reveal their contributions in structural evolution, we further prepared PVA/PAA (PA) samples without ionic crosslinking and non-salting-out treatment (unsoaked in the Na$_2$SO$_4$/Gly/H$_2$O ternary solvent), and S-PVA/PAA (S-PA) samples with salting-out treatment only for subsequent studies. As shown by scanning electron microscope (SEM), with the step-by-step introduction of ionic coordination and crystalline domain crosslinking, the network structure of hydrogel fibers was gradually denser, implying the synergistic enhancement of molecular (chain) interactions by ionic coordination and crystalline domain

crosslinking (Supplementary Fig. 1). In the fourier-transform infrared (FTIR) spectra, the -OH characteristic peak of the PA gradually shifted from 3369 cm$^{-1}$ to 3322 cm$^{-1}$ with the introduction of Zr$^{4+}$ and salting-out treatment, implying their synergistic enhanced the hydrogen bonding interactions in the S-PAZr (Fig. 2a). In addition, the characteristic peaks of -COO$^-$ at ~1575 cm$^{-1}$ and the characteristic peaks in 850–950 cm$^{-1}$ could be attributed to the interaction of -COO$^-$ with ions and the introduction of the ternary solvent, respectively, which was reconfirmed by high-resolution O 1s XPS spectra (Fig. 2b)[37,38]. Notably, the relative number of Zr-O bonds in the PAZr decreased from 10.31% to 5.53% after salting-out treatment, while the binding energy of the S-PAZr exhibited an unremarkable shift compared with that of the PAZr in high-resolution Zr 3d XPS spectra. This indicated that the number of Zr$^{4+}$ crosslinking in the S-PAZr was reduced after the salting-out treatment, whereas the Zr$^{4+}$ coordination state was not significantly changed (Fig. 2c)[39]. The crystalline domain evolution of the S-PAZr was discussed by X-ray diffraction (XRD) analysis. As shown in Fig. 2d, the non-salting-out treated PA and PAZr exhibited mainly H$_2$O diffraction peaks and absence of obvious diffraction peaks on the typical (101) reflective crystal plane of PVA. In contrast, the S-PA and S-PAZr displayed diffraction peaks at $2\theta$ = 19.5° due to the Hofmeister effect and water outflow during the salting-out treatment (Supplementary Fig. 2)[40–42]. Further, the crystallinity of the S-PA and S-PAZr were roughly assessed to be 6.01% and 5.58% by calculating the ratio of the (101) crystal reflected area at $2\theta$ = 18–21° to the relative area of the entire diffraction peak (Fig. 2e)[43,44]. Apparently, the salting-out treatment was beneficial in increasing the crystallinity of the sample, and the slight decrease in the crystallinity of the S-PAZr could be

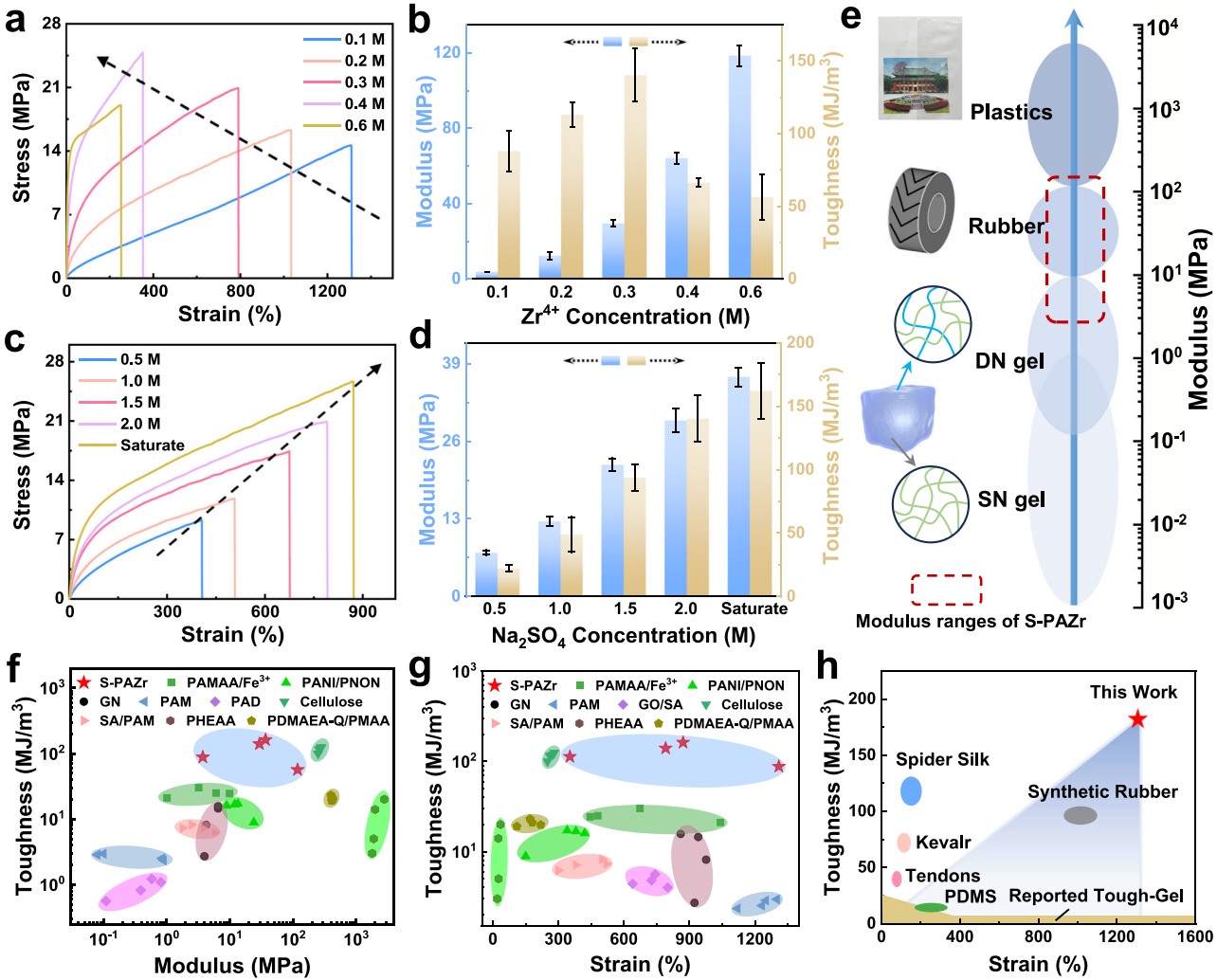

**Fig. 3 | Mechanical properties of the S-PAZr hydrogel fiber.** Effect of (**a**, **b**) $Zr^{4+}$ and (**c**, **d**) $Na_2SO_4$ concentration on the mechanical properties of S-PAZr hydrogel fibers. Data were presented as mean ± SD ($n = 3$ independent samples). **e** Modulus ranges of gels, rubbers, plastics, and S-PAZr hydrogel fibers. Ashby plots of (**f**) toughness versus elastic modulus, (**g**) toughness versus tensile strain of S-PAZr hydrogel fibers, and other reported hydrogel fibers. **h** Mechanical performance comparison of S-PAZr hydrogel fibers and other high-toughness materials. The blue shaded area showed the range of toughness and strain that could be adjusted by the BSE strategy.

related to the salt-in effect caused by $ZrOCl_2$[45]. In summary, the contribution of introducing ionic crosslinking and crystalline domains to the structural evolution of the S-PAZr was revealed in detail. First, the direct introduction of ionic crosslinks reinforced the hydrogen bonding interactions of the hydrogel fiber and formed additional ionic crosslinking sites. Further, the ionic crosslinked hydrogel fiber exhibited a complex change during the salting-out treatment, including outflow of $H_2O$ and $Zr^{4+}$, inflow of $Na^+$, $SO_4^{2-}$, and Gly, as well as formation of new crystalline domains (Fig. 2e inset). These behaviors contribute to stronger hydrogen bonding interactions, suitable ionic crosslinking levels, and extra crystalline domains with high energy dissipation capacity in hydrogel fibers.

The toughening mechanism of S-PAZr was further revealed by SEM and small-angle X-ray scattering (SAXS) tests. As shown in Supplementary Fig. 3, the network structure of S-PAZr would be oriented during stretching, which was also confirmed by the change of the 2D SAXS pattern from a circle to an ellipse (Fig. 2f inset). In the 1D SAXS curves, the scattering of the S-PAZr at 0.016 Å$^{-1}$ and 0.082 Å$^{-1}$ were mainly attributed to the interaction of PAA with cations and the semicrystalline lamellar domains of PVA, respectively[46–48]. When the S-PAZr was stretched from 0% to 500% strain, the scattering at 0.016 Å$^{-1}$ and 0.082 Å$^{-1}$ were shifted to 0.015 Å$^{-1}$ and 0.078 Å$^{-1}$,

respectively. It was implied that the carboxylic acid-cation interactions and PVA crystalline domains were damaged for dissipating energy during the stretching (Fig. 2g). Benefiting from the elaborated ionic crosslinking and crystalline domains, the S-PAZr hydrogel fiber could synergistically dissipate energy through ionic decoordination and crystalline domain unfolding during stretching (Supplementary Figs. 4, 5)[39,49–51]. As a result, the S-PAZr hydrogel fiber exhibited high mechanical properties, such as tensile stress of 24.43 ± 2.11 MPa, tensile strain of 844.44 ± 107.21%, elastic modulus of 36.81 ± 1.58 MPa, and toughness of 162.25 ± 21.99 MJ/m$^3$ (Fig. 2h). Figure 2i demonstrated the significant positive effects of constructing ionic crosslinking and crystalline domains on the mechanical properties of the PA hydrogel fiber. It was observed that the tensile stress, elastic modulus, and toughness of the S-PAZr hydrogel fiber were 30924.05%, 87642.86%, and 42697.37% higher than those of the PA hydrogel fiber, respectively.

## Broadly adjustable mechanical properties of the S-PAZr hydrogel fiber through ionic coordination and Hofmeister effects

The mechanical properties of S-PAZr hydrogel fibers could be easily adjusted by altering the concentrations of PVA, AA, $Zr^{4+}$, and $Na_2SO_4$, as well as the salting-out treatment time (Supplementary Fig. 6 and

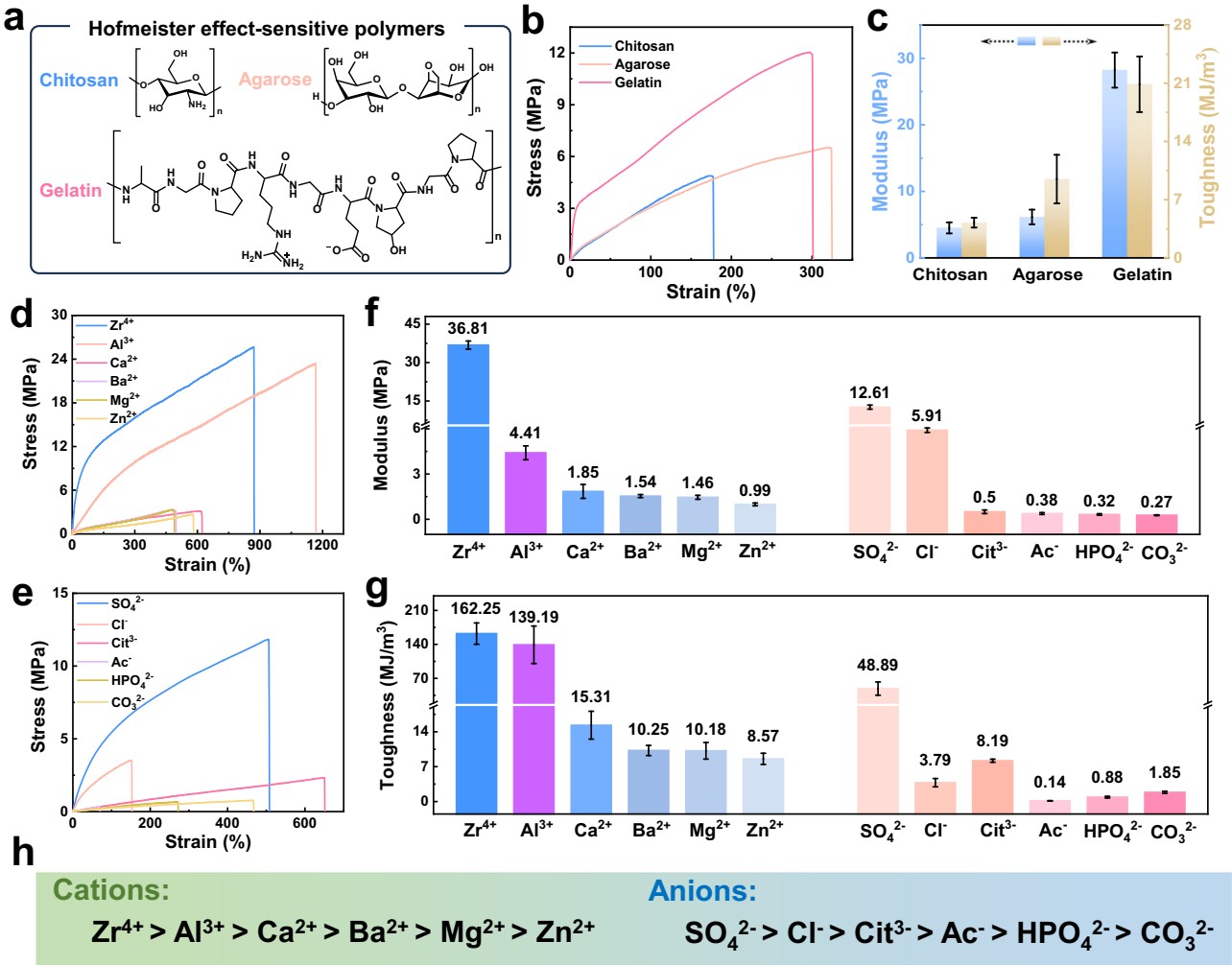

**Fig. 4 | The BSE strategy was generalized to different combinations of Hofmeister effect-sensitive polymers and inorganic salts. a** Hofmeister effect-sensitive polymers such as chitosan, agarose, and gelatin. **b, c** Mechanical properties of bionic hydrogel fibers prepared with 1 wt% chitosan, 2 wt% agarose, and 10 wt% gelatin, respectively. Data were presented as mean ± SD (*n* = 3 independent samples). **d** Stress-strain curves of bionic hydrogel fibers with different 0.3 M cationic crosslinking and saturated Na₂SO₄-based ternary solvent treatment. **e** Stress-strain curves of bionic hydrogel fibers with 0.3 M Zr⁴⁺ crosslinking and different 1 M sodium salts-based ternary solvent treatment. **f** Modulus and (**g**) toughness of bionic hydrogel fibers tuned by various ions. Data were presented as mean ± SD (*n* = 3 independent samples). **h** According to the elastic modulus of bionic hydrogel fibers, the effect sequence of ions on the mechanical properties of bionic hydrogel fibers.

Fig. 3a–d). In particular, the concentrations of Zr⁴⁺ and Na₂SO₄, which were directly related to ionic crosslinking and crystalline domains. With the Zr⁴⁺ concentration increased from 0.1 to 0.6 M, the elastic modulus of the S-PAZr hydrogel fiber improved remarkably from gel to polyethylene levels (3.74 ± 0.16–118.53 ± 5.49 MPa) due to the higher ionic crosslinking density (Fig. 3e). Whereas, the increase in ionic crosslinking density also led to the decrease in tensile strain of S-PAZr hydrogel fibers. Unlike ionic crosslinking, the PVA crystalline domains induced by the Hofmeister effect were adaptive in that they were able to restrict the polymer network deformation at the small strain and gradually unfolding to achieve sustained energy dissipation during further deformation[51,52]. Therefore, S-PAZr hydrogel fibers exhibited an overall improvement in tensile stress, tensile strain, elastic modulus, and toughness with increasing Na₂SO₄ concentration. Taken together, the mechanical properties of S-PAZr hydrogel fibers could be easily customized over a broad range, including tensile stress from 8.63 ± 0.99 to 24.43 ± 2.11 MPa, tensile strain from 40.56 ± 16.24 to 1155.74 ± 165.42%, elastic modulus from 3.74 ± 0.16 to 118.53 ± 5.49 MPa, and toughness from 4.82 ± 2.57 to 162.25 ± 21.99 MJ/m³. In addition, we visually

compared the mechanical properties of S-PAZr hydrogel fibers with other reported hydrogel fibers and common high-toughness materials (Fig. 3f–h). The toughness of the S-PAZr was higher than that of most current hydrogel fibers and well those of anhydrous polymers such as polydimethylsiloxane (PDMS), Kevlar, and synthetic rubber, even exceeding the toughness of natural tendon and spider silk[11,18,25,36,53–58].

Further, different Hofmeister effect-sensitive polymers and inorganic salts were adopted to prepare bionic hydrogel fibers for investigating the universality of the BSE strategy. As expected, chitosan, agarose, and gelatin were typical Hofmeister effect-sensitive polymers, which were involved in the construction of bionic hydrogel fibers with higher mechanical properties than the S-PAA/Zr⁴⁺ hydrogel fiber without Hofmeister effect-sensitive polymers (Fig. 4a–c and Supplementary Fig. 7). For inorganic salts, they mainly regulated the ionic crosslinking and crystalline domains of bionic hydrogel fibers through the coordination effect of cations and the Hofmeister effect of anions to adjust the mechanical properties of bionic hydrogel fibers. Therefore, multivalent cations and anions with salting-out effects were used to discuss the toughening effect of the BSE strategy on hydrogel fibers,

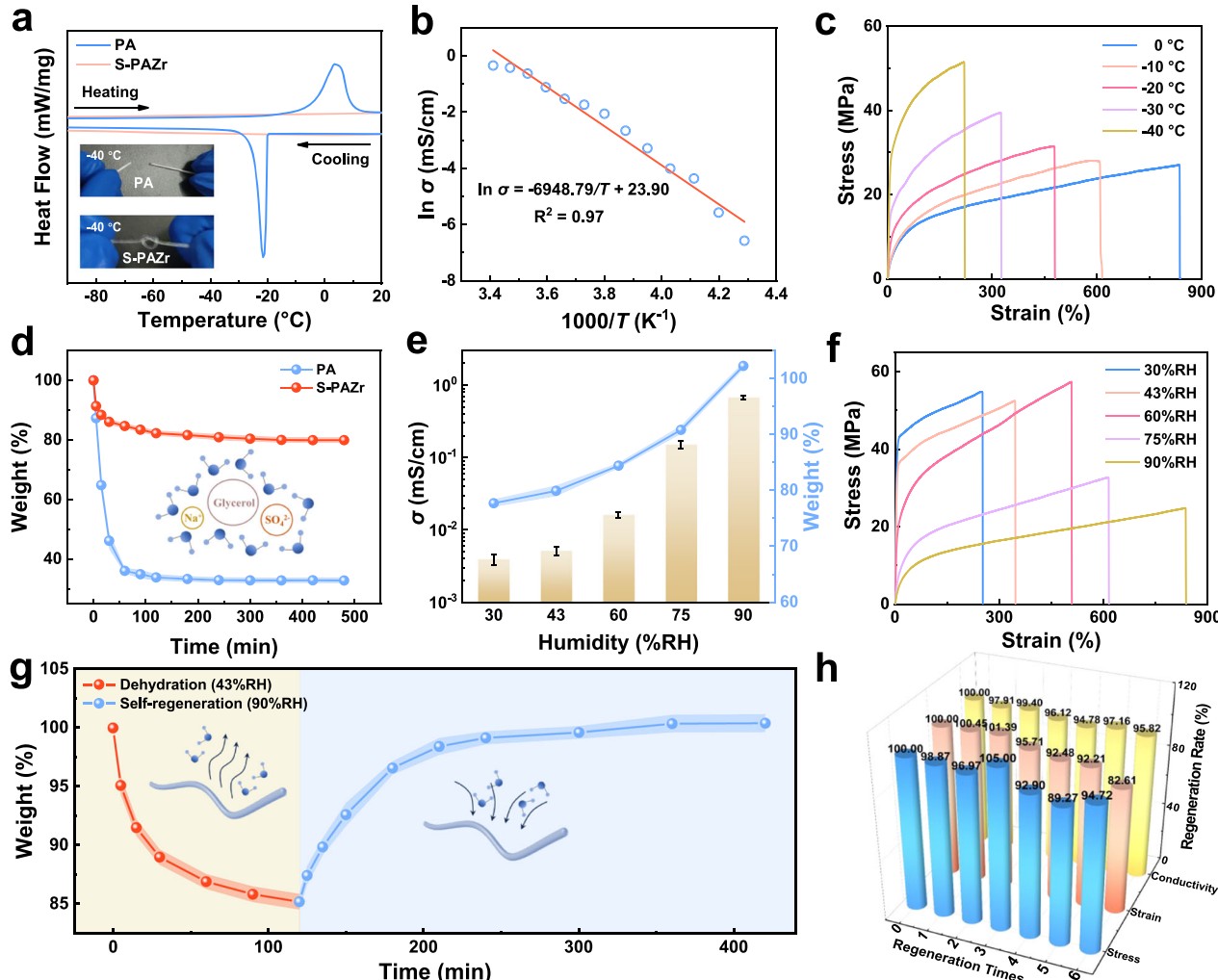

**Fig. 5 | Environmental tolerance of the S-PAZr hydrogel fiber prepared by the BSE strategy. a** DSC analysis of PA and S-PAZr hydrogel fibers. The inset showed PA and S-PAZr hydrogel fibers at −40 °C. **b** Electrical and (**c**) mechanical properties of the S-PAZr hydrogel fiber with decreasing temperature. **d** Weight loss of PA and S-PAZr hydrogel fibers at 43%RH. The inset showed the schematic interaction of Gly and Na$_2$SO$_4$ with H$_2$O. The ball-and-stick model in the inset was the water molecule. **e** The weight, conductivity, and (**f**) mechanical properties of S-PAZr hydrogel fibers at different humidity. Data were presented as mean ± SD ($n$ = 3 independent samples). **g** The dehydrated S-PAZr hydrogel fibers self-regenerate to their initial weight. The ball-and-stick model in the inset was the water molecule and the fiber represented the S-PAZr hydrogel fiber. **h** Electrical and mechanical properties of self-regenerated S-PAZr hydrogel fibers.

respectively. As shown in Fig. 4d, bionic hydrogel fibers crosslinked with divalent cations such as Ca$^{2+}$, Ba$^{2+}$, Mg$^{2+}$, and Zn$^{2+}$ exhibited tensile stress > 2.5 MPa and tensile strain > 450%, while bionic hydrogel fibers crosslinked with high-valent cations such as Zr$^{4+}$ and Al$^{3+}$ showed higher mechanical properties (tensile stress > 20 MPa, tensile strain > 850%) owing to the more stable ionic coordination[59]. It was implied that the high-valent cations with strong coordination ability were more favorable to synergize the Hofmeister effect for achieving hydrogel fibers with high mechanical properties. Figure 4e presented the mechanical properties of bionic hydrogel fibers obtained by using different anionic for salting-out treatments. The bionic hydrogel fibers treated with Na$_2$SO$_4$ salting-out treatments showed a stress of > 10 MPa and a strain of > 500%, whereas the toughening of the bionic hydrogel fibers salting-out treated with other sodium salts with different anions was not as effective. Nevertheless, the bionic hydrogel fiber with the weakest mechanical properties, the bionic hydrogel fiber salting-out treated with sodium acetate, still displayed a stress of > 0.35 MPa and a strain of > 60%. According to the elastic modulus of bionic hydrogel fibers, the sequence of anionic effects on the bionic hydrogel fibers was SO$_4^{2-}$ > Cl$^-$ > Cit$^{3-}$ > Ac$^-$ > HPO$_4^{2-}$ > CO$_3^{2-}$, which was inconsistent with the typical Hofmeister sequence (Fig. 4f–h). The insignificant

toughening effect of the Cit$^{3-}$, Ac$^-$, HPO$_4^{2-}$, and CO$_3^{2-}$ anions could be attributed to the alkaline environment caused by them, in which the coordination of Zr$^{4+}$ with -COO$^-$ in hydrogel fibers was broken and hydrogel fibers underwent swelling (Supplementary Fig. 8)[60]. Evidently, the toughening effect of the BSE strategy on hydrogel fibers was universal for different combinations of Hofmeister effect-sensitive polymers and inorganic salts. The excellent toughening effect of the BSE strategy required the synergy of robust coordinated cations and kosmotropic salts with less damage to ionic coordination.

## Environmental tolerance of the S-PAZr hydrogel fiber

The S-PAZr hydrogel fiber with excellent mechanical properties was used as a representation of bionic hydrogel fibers to exhibit their environmental tolerance. As shown in Fig. 5a inset, the PA hydrogel fiber was frozen and brittle at −40 °C due to the high percentage of free water, while the S-PAZr hydrogel fiber benefited from the multiple interactions of the polymer chains, Gly, inorganic salts, and H$_2$O, and it remained transparent and withstood the knotting treatment at −40 °C. The differential scanning calorimetry (DSC) analysis further compared their anti-freezing properties. Compared to the PA hydrogel fiber, the S-PAZr hydrogel fiber lacked significant crystallization and

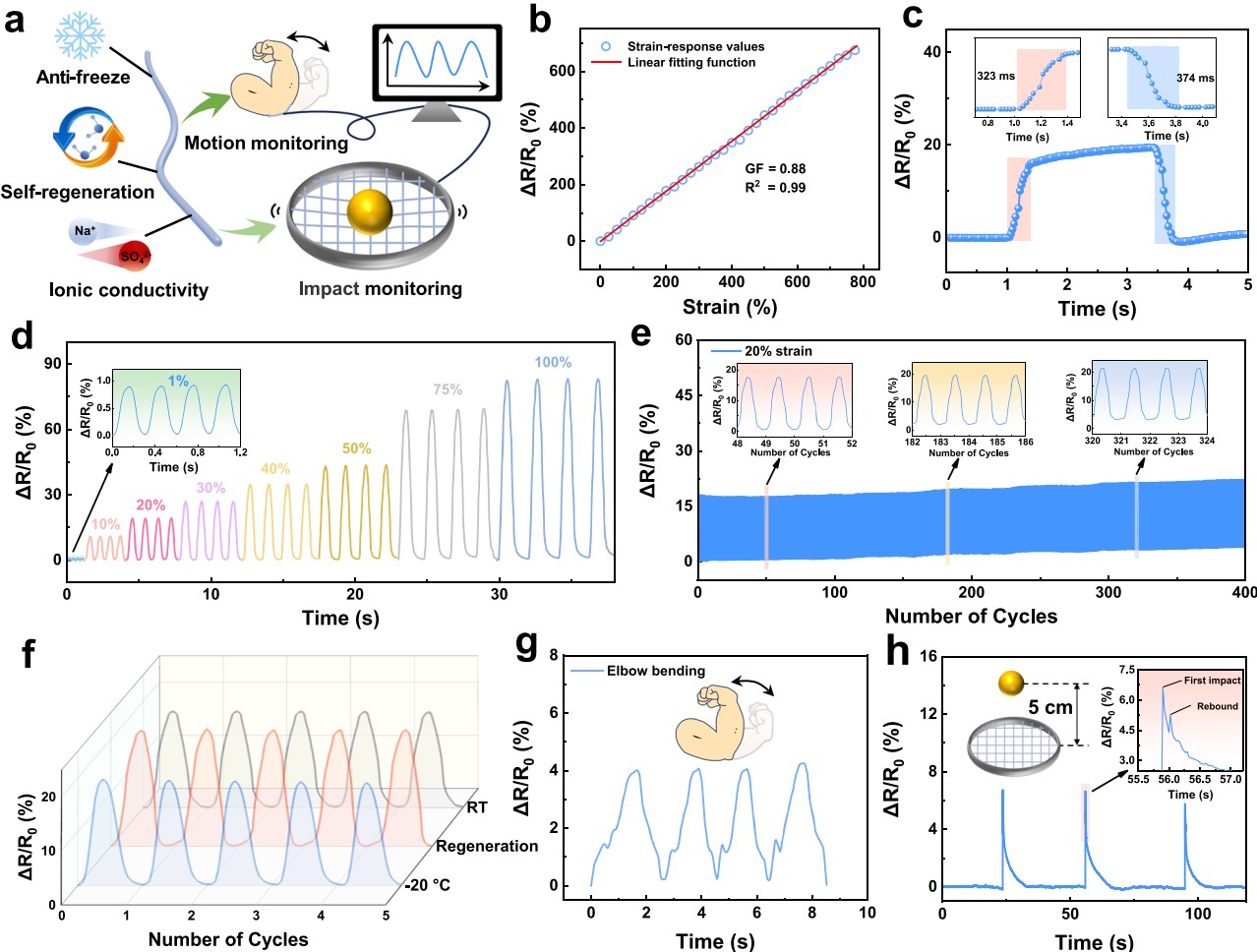

**Fig. 6 | Flexible sensing performances of the S-PAZr hydrogel fiber. a** The advantages and flexible sensing applications of S-PAZr hydrogel fiber. **b** Sensitivity and (**c**) response/recovery time of S-PAZr hydrogel fiber sensor. **d** The response of S-PAZr hydrogel fiber sensor at different strains. **e** The S-PAZr hydrogel fiber sensor at 20% strain for 400 cycles. **f** 20% strain cycling test of S-PAZr hydrogel fiber sensor at RT, −20 °C, and after self-regeneration. The S-PAZr hydrogel fiber sensor for (**g**) motion and (**h**) impact monitoring.

melting peaks in the range of −100 to 20 °C, implying the S-PAZr hydrogel fiber with great anti-freezing properties[61]. The conductivity of the S-PAZr hydrogel fiber at low temperatures was shown in Fig. 5b. Owing to the low-temperature stability of the S-PAZr hydrogel fiber, it had a conductivity of 0.0014 mS/cm at −40 °C, and the conductivity-temperature relationship followed the Arrhenius equation. In addition, the S-PAZr hydrogel fiber gradually hardened with decreasing temperature, but it still exhibited a strain of > 200% at −40 °C (Fig. 5c).

The strong hydrogen bonding effect of Gly and the hydration effect of inorganic salts endowed S-PAZr hydrogel fibers with the water retention capacity without any coating protection (Fig. 5d inset). As shown in Fig. 5d, the common PA hydrogel fibers rapidly lost water to 32.78% of the initial weight at 43%RH, while the S-PAZr hydrogel fibers still maintained 79.91% of the initial weight. On the other hand, the water content, electrical and mechanical properties of S-PAZr hydrogel fibers could be adjusted by the ambient humidity. As expected, the increase in ambient humidity allowed S-PAZr hydrogel fibers to exhibit higher water content, better conductivity, lower elastic modulus, and longer strain (Fig. 5e, f and Supplementary Fig. 9). Notably, the S-PAZr hydrogel fiber exhibited 102.26 ± 0.67% of weight at 90%RH, implying that it could spontaneously capture water molecules from the air at high humidity. Based on the hygroscopicity of the S-PAZr hydrogel fiber, the dehydrated S-PAZr hydrogel fiber could be self-regenerated

by resting at 90%RH for recovering its initial weight as well as the corresponding electrical and mechanical properties. The electrical and mechanical properties of S-PAZr hydrogel fibers were almost completely restored during the first three self-regenerations (Fig. 5g, h). It was obvious that the excellent environmental tolerance of S-PAZr hydrogel fibers prepared by the BSE strategy was friendly for both their storage and application.

### Flexible sensing application of the S-PAZr hydrogel fiber
Benefiting from the excellent mechanical properties, environmental tolerance, and ionic conductivity of the S-PAZr hydrogel fiber, it could be used for flexible sensing applications such as motion and impact monitoring (Fig. 6a). As shown in Fig. 6b, stretching the hydrogel fibers narrowed the ion migration channels while lengthening the migration paths, resulting in an improvement in the resistance of the hydrogel fibers with increasing tensile strain. Therefore, a correlation was established between the relative resistance change ($\Delta R/R_0$) and strain of the S-PAZr hydrogel fiber sensor, which exhibited a linear relationship with a gauge factor (GF) of 0.88. In addition, the S-PAZr hydrogel fiber sensor was sensitive with a response time of 323 ms and recovery time of 374 ms (Fig. 6c). More importantly, the sensor provided a highly stable and reliable resistance signal. When it was repeatedly stretched to the same strain, its $\Delta R/R_0$ was similar (Fig. 6d). And the $\Delta R/R_0$ remained stable after 400 cycles of 20% strain (Fig. 6e). The

environmental tolerance of the S-PAZr hydrogel fiber allowed it to work at low temperature and after self-regeneration (Fig. 6f). The $\Delta R/R_0$ of the S-PAZr hydrogel fiber sensor at −20 °C and after self-regeneration was similar to that measured at room temperature (RT). Figure 6g demonstrated the monitoring of elbow bending by using the S-PAZr hydrogel fiber sensor, where multiple bends of the elbow could be stably monitored. Furthermore, impact monitoring could be achieved by weaving S-PAZr hydrogel fibers into a tennis racket (Fig. 6h). The impact of the ball relaxed the conductive pathways that were originally in close contact, resulting in an increase in resistance. Interestingly, the secondary impact caused by the rebound of the ball was also monitored, implying that the S-PAZr hydrogel fiber sensor was sensitive and reliable.

## Discussion

Inspired by the multilevel adjustment of spider silk network structure by ions, we proposed a BSE strategy for fabricating hydrogel fibers with high mechanical properties and environmental tolerance. The bionic hydrogel fibers prepared by the BSE strategy and the improved self-lubricating spinning strategy, which dissipated energy via the synergistic of ionic crosslinking and crystalline domains, exhibiting a toughness of $162.25 \pm 21.99$ MJ/m$^3$ comparable to that of spider silk. In addition, bionic hydrogel fibers allowed easily customize of their mechanical properties by regulating the ionic crosslinking and crystalline domains, especially the elastic modulus could be raised from gel to plastic levels ($0.27 \pm 0.025$–$118.53 \pm 5.49$ MPa). Benefiting from the strong hydrogen bonding of Gly and the hydration effect of inorganic salts, bionic hydrogel fibers exhibited conductivity of 0.0014 mS/cm and excellent mechanical properties (stress > 50 MPa, strain > 200%) at −40 °C, as well as spontaneously capturing water molecules to self-regenerate to their original state after dehydration. More importantly, the proposed BSE strategy was found to be generalizable to different combinations of Hofmeister effect-sensitive polymers and inorganic salts. Considering the prevalence of ionic coordination and Hofmeister effects in various polymer and solvent systems, we were confident that the BSE strategy could be applied to more systems to promote their applications.

## Methods

### Materials

PVA (PVA-1799), acrylic acid (AA), 2-hydroxy-2-methylpropiophenone (I1173), ZrOCl$_2$·8H$_2$O, chitosan (deacetylation degree ≥ 95%, viscosity is 100–200 mPa·s), and ZnCl$_2$ were purchased from Shanghai Macklin Biochemical Co., Ltd. Agarose (high gel strength) and gelatin (gel strength ~250 g Bloom) were purchased from Shanghai Aladdin Bio-Chem Technology Co., Ltd. AlCl$_3$ was purchased from Sinopharm Chemical Reagent Co., Ltd. Gly, K$_2$CO$_3$, NaBr, KCl, CaCl$_2$, BaCl$_2$, MgCl$_2$, Na$_2$SO$_4$, NaCl, Na$_3$Cit, NaAc, Na$_2$HPO$_4$, and Na$_2$CO$_3$ were purchased from Guangzhou Chemical Reagent Factory. The deionized water was obtained using a water purification system.

### The improved self-lubricating spinning strategy

We previously reported a self-lubricating spinning strategy based on hydrophobic substrate-induced regional heterogeneous polymerization, which enabled the fabrication of hydrogel fibers from monomers and allowed the researcher to freely design the network structure of hydrogel fibers[36]. In practice, we found that the self-lubricating spinning strategy may be limited by residual oxygen. As a result, continuous spinning of hydrogel fibers over time was prone to clogging and it was difficult to obtain long hydrogel fibers (> 10 m).

To solve the above problems, we have improved the self-lubricating spinning strategy. Regional heterogeneous polymerization induced by the hydrophobic substrate would lead to the formation of lubricating liquid and gel fibers. Notably, gel fibers with diameters smaller than the inner diameter of the FEP tube. Therefore,

highly polymerized gel fibers could be easily pushed out with the assistance of the lubricating liquid. Here, we only push out a part of the gel fiber and another part remained in the FEP tube. Then, after waiting for the subsequent spinning solution to infiltrate the gel fibers remaining in the FEP tube, UV polymerization was performed to form longer gel fibers. Repeating the process of partial push-out of hydrogel fibers and UV polymerization allowed continuous spinning of hydrogel fibers.

### Fabrication of S-PAZr hydrogel fibers by the BSE strategy and the improved self-lubricating spinning strategy

First, 1 g PVA, 2 g AA, 0.96 g ZrOCl$_2$·8H$_2$O, and photoinitiator (I1173, 1 mol% of monomer) were H$_2$O-sized to 10 mL as the spinning solution. The spinning solution was injected by a syringe pump into the fluorinated ethylene propylene (FEP) tube with 1 mm inner diameter. The spinning solution in the FEP tube was irradiated by a 365 nm UV light with a power of 18 W for initiating free radical polymerization to achieve the PAZr hydrogel fiber. The UV irradiation time was controlled to be 15 min. Then, the PAZr hydrogel fiber was washed in a water bath to remove residual monomers and immersed in a Gly/H$_2$O (weight ratio of 1 : 4, which exhibited a high solubility of Na$_2$SO$_4$) solution containing saturated Na$_2$SO$_4$ for 24 h for conversion to the S-PAZr hydrogel fiber.

### The universality of the BSE strategy

For bionic hydrogel fibers with different Hofmeister effect-sensitive polymers, chitosan, agarose, and gelatin-based bionic hydrogel fibers were 1 wt%, 2 wt%, and 10 wt%, respectively. The other parameters were consistent with the fabrication of the S-PAZr hydrogel fiber.

For bionic hydrogel fibers with different cationic coordination, bionic hydrogel fibers replaced ZrOCl$_2$ with inorganic salts containing coordinatable cations (AlCl$_3$, CaCl$_2$, BaCl$_2$, MgCl$_2$, and ZnCl$_2$). The other parameters were consistent with the fabrication of the S-PAZr hydrogel fiber.

For bionic hydrogel fibers obtained by using different anionic for salting-out treatments, bionic hydrogel fibers were achieved by immersing Gly/H$_2$O (mass ratio of 1 : 4) solutions containing 1 M Na$_2$SO$_4$, NaCl, Na$_3$Cit, NaAc, Na$_2$HPO$_4$, and Na$_2$CO$_3$ for 24 h. The other parameters were consistent with the fabrication of the S-PAZr hydrogel fiber.

### Characterizations

FTIR tests of the samples were acquired using a Nicolet IS50-Nicolet Continuum produced by Thermo Fisher Scientific, Ltd., where attenuated total reflection (ATR) mode was employed for sample detection. XPS analysis was performed on a Kratos Axis Ultra-DLD X-ray photoelectron spectrometer. The XRD test was performed on the PANalytical (X'Pert PRO) instrument with an X-ray of $\lambda = 0.154$ nm, operated at 40 kV and 40 mA, and the XRD profiles were collected in the $2\theta$ range of 5°–60°. The SAXS measurements were carried out at Xeuss 3.0 SAXS system (Xenocs, France) with an X-ray of $\lambda = 0.154$ nm. The sample-to-detector distance was 1500 mm, and the exposure time was set as 600 s. The anti-freezing properties of hydrogel fibers were measured on DSC (NETZSCH 3500 Sirius) with a heating/cooling rate of 10 °C/min under nitrogen flow. All hydrogel samples were soaked in deionized water for 2 h for removing as much Gly as possible to complete freeze-drying. The freeze-dried hydrogels were brittlely exposed to cross sections with liquid nitrogen and sputtered with gold, followed by SEM (HITACHI SU8220) test.

### Tensile tests

The mechanical properties of hydrogel fibers were tested by using a universal tester (AGS-X) with an environmental chamber and a liquid nitrogen tank. The tensile speed was fixed at 100 mm/min. The constant temperature time for low-temperature tensile test was 30 min.

## Conductivity tests

The electrical resistance of hydrogel fiber was measured by using a SourceMeter (Keithley DMM7510). The conductivity ($\sigma$) was calculated as $\sigma = \Delta l/(R \times S)$, where $\Delta l$, $R$, and $S$ were the length, electrical resistance, and cross-sectional area of hydrogel fiber, respectively.

## Sensing tests of hydrogel fiber sensors

The sensing tests of the hydrogel fiber sensor were performed by the combination of a universal tensile testing machine (AGS-X) and a SourceMeter (Keithley DMM7510). The gauge factor (GF) was estimated as GF = $\Delta R/R_O/\varepsilon$, where $\Delta R$ was the resistance changes with strain, $R_O$ was the resistance of the hydrogel fiber at the original length, and $\varepsilon$ was the applied strain.

## Statistics and reproducibility

All experiments were repeated independently with similar results for at least three times. All data were expressed as the mean ± standard deviation (SD).

## Data availability

The data that support the findings of this study are included in the published article (and its Supplementary Information) or available from the corresponding author on request.

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

## Acknowledgements

This study was supported by the National Natural Science Foundation of China (51873074 (Y.Y.)).

## Author contributions

S.W., Y.Y., and W.F. conceived the research; S.W., Y.Y., W.L., C.G., W.F., and Z.Q. designed the experiments; S.W., Z.L., S.X., and R.W. performed the experiments; S.W., Z.L., S.X., and R.W. designed, fabricated, and characterized composites; S.W., Y.Y., W.L., and C.G. interpreted the data, analyzed the data and wrote the manuscript. All authors discussed the data and direction of the project at regular intervals throughout the study.

## Competing interests

The authors declare no competing interests.
