## [Peer Review File · Nature Communications]

Spider-silk-inspired strong and tough hydrogel fiber with anti-freezing and water retention propertiesREVIEWER COMMENTS

Reviewer #1 (Remarks to the Author):

In this manuscript, strong and tough hydrogel fibers were developed by utilizing the synergistic effect of ionic cross-linking (using high valent cation) and salting out (using kosmotropic anions). The generality of this strategy was established with a combination of different Hofmeister effect sensitive polymers and inorganic salts. Solvent exchange with glycerol rendered anti-freezing properties to the hydrogel fibers. The approach is systematic and the quality of the presentation in the manuscript is very good. However, the following points should be addressed before the manuscript can be accepted for publication.

1. How were the tensile stress-strain plots in Figure 5 obtained at low temperature? The details of the experimental set up should be shared, along with a photograph.
2. The strength of the fibers may change depending on the draw ratio of the fibers. The PVA chains may also get aligned during the spinning process, which may have an additional effect in the observed mechanical properties. These aspects should be discussed in the manuscript.
3. Is there any difference in microstructure of the hydrogel fibers (PA/PVA) when prepared in presence/absence of the metal ions and in presence/absence of salting out effect? SEM analysis may shed some light on this aspect.
4. SAXS experiments analyse a small section of the sample. How homogeneous are the hydrogel fibers? What is the mean and standard deviation for the values provided for the distance between crystalline domains and the estimated radius of these domains?
5. The SAXS scattering data have been compared at 0% and 300% strain. At least one more data set should be provided at higher strain% to confirm the trend.
6. There are numbering mistakes in Figure 4, which should be corrected.

Reviewer #2 (Remarks to the Author):

In this study, inspired by the structure of spider silk, the authors constructed a biomimetic hydrogel fiber with high toughness and mechanical strength. The hydrogel is capable of adapting to low temperatures and dry environments, and is able to regenerate itself after water loss without loss of mechanical and electrochemical properties. The authors also made reasonable experiments to verify this conclusion. Therefore, the manuscript is recommended for publishing in Nature Communications after minor revisions as follows:

1. The introduction states that "it remains a challenge to achieve both high mechanical properties and environmental adaptability of hydrogel fibers.". However, the manuscript only demonstrates that the bionic hydrogel fibers have good performance at -40 °C and in dry environments. It is therefore recommended that more precise wording be used.
2. How did the authors determine the ratio of the inorganic salt/glycerol/H₂O ternary solvent? This doesn't seem to come across in the experimental section.
3. It is suggested that on page 5, line 125, the words "Na₂SO₄/glycerol/H₂O" should be replaced by

"Na₂SO₄/Gly/H₂O" for consistency with the text.

4. In Figure 4, the font of "Hofmeister effect" is wrong; on page 16, the format of "Δ" is not standardized; in Figure S6, the font of horizontal coordinates is not standardized. There are also some formatting errors, which the authors are also asked to double-check and revise.

5. In the introduction section, it is recommended to provide a clearer and more comprehensive description of the background, purpose and significance of the study. It is also recommended to refer to some literature on hydrogel sensing and the Hofmeister effect (e.g., *Adv. Mater.* 2021, 33(11): 2007829; *Adv. Funct. Mater.* 2023, 33(12): 2213283; *Nat. Commun.* 2017, 8(1): 15911).

6. On page 12 it says "The superior toughening effect of the BSE strategy requires the synergy of robust coordinated cations with kosmotropic salts...". How is this synergy achieved? Does it have any effect on the various properties of hydrogel fibers?

7. What is the lifetime of this hydrogel fiber? Is there a limit to the number of self-regenerations? Do other external environmental factors affect the performance of the material? How did the authors avoid these problems?

8. It is recommended that the format of references be standardized, e.g., some authors are fully marked out and some are omitted. It is recommended that the full text be revised in accordance with the journal format.

Reviewer #3 (Remarks to the Author):

In the manuscript entitled "A Strong and Tough Hydrogel Fiber with Environmental Adaptability Inspired by Spider Silks", authors constructed a hydrogel fiber with elaborated ionic crosslinking and crystalline domains inspired by the multilevel adjustment of spider silk network structure with ions. In addition, the introduction of inorganic salt/glycerol/H₂O ternary solvent allowed these hydrogel fibers to adapt to low-temperature and dry environments with almost no loss of mechanical and electrical properties. Furthermore, this work provided ideas to fabricate hydrogel fibers for flexible sensing applications. However, there are still many problems to be solved. Therefore, the referee didn't recommend publishing it in *Nature Communications*.

1. Despite the strong mechanical properties of the hydrogel fibers, the sensing performance is not outstanding when it is used as sensing application.
2. The authors should be strict about some terms, such as the description of relative intensity in XPS.
3. Compared with other works (*Nature Communications*, 2022, 13:3408; *ACS Appl. Mater. Interfaces*, 2020, 12, 25353–25362), the hydrogel fibers have low conductivity at low temperature.
4. For the part of structural evolution and toughness mechanism, more characterization is needed to verify its accuracy, such as SEM, TEM and more.

Response to the Comments from the Reviewers:

We thank reviewers for their valuable comments and helpful suggestions. Consequently, we have carefully addressed all concerns raised by the reviewers. The major revision was highlighted in blue in the manuscript and supplementary information. Below, we outline point-by-point responses to address their comments and revise some grammar and sentences, all of which are very helpful for further strengthening the manuscript.

REVIEWER COMMENTS

Reviewer #1 (Remarks to the Author):

In this manuscript, strong and tough hydrogel fibers were developed by utilizing the synergistic effect of ionic cross-linking (using high valent cation) and salting out (using kosmotropic anions). The generality of this strategy was established with a combination of different hofmeister effect sensitive polymers and inorganic salts. Solvent exchange with glycerol rendered anti-freezing properties to the hydrogel fibers. The approach is systematic and the quality of the presentation in the manuscript is very good. However, the following points should be addressed before the manuscript can be accepted for publication.

Response: We appreciate this reviewer for the thorough review and positive comments on our work.

1. How were the tensile stress-strain plots in Figure 5 obtained at low temperature? The details of the experimental set up should be shared, along with a photograph.

Response: The tensile stress-strain diagrams in Figure 5 were obtained at low temperatures by using a universal tester (AGS-X, Shimadzu) with an environmental chamber and a liquid nitrogen tank. Photographs and details of the experimental setup were shown in Figure R1. During the low temperature tensile test, liquid nitrogen was pumped from the liquid nitrogen tank into the environmental chamber with a control program to achieve a low temperature

environment. Further, the collection of tensile data was realized by the universal tester after being kept in a low temperature environment for 30 min.

Figure R1. (a) Front and (b) side views of the experimental setup for performing low temperature tensile tests (as requested by the editor, the parts involving laboratory information have been obscured).

2. The strength of the fibers may change depending on the draw ratio of the fibers. The PVA chains may also get aligned during the spinning process, which may have an additional effect in the observed mechanical properties. These aspects should be discussed in the manuscript.

Response: Great point. The strength of our constructed bionic hydrogel fibers did change depending on the draw ratio, as verified in our ongoing follow-up research. The tensile stress of the bionic hydrogel fibers was improved by about 1-3 times with increasing draw ratio, while the tensile strain decreased significantly.

We agree with the reviewer that if PVA chain segments underwent orientational alignment during the spinning process, it would generate additional effects on the mechanical properties of bionic hydrogel fibers. Differential flow speeds in the tube could lead to the orientation of the PVA chain segments, but we used a slow feed speed and a tube with a constant diameter of 1 mm, the orientation and de-orientation of PVA chain segments were simultaneous in the spinning process. In fact, the orientation of PVA chain

segments was not clearly observed in our work, which was also confirmed by the supplementary SEM, EDS and SAXS tests (Figure R2). We thank the reviewer for the enlightening question, and in our subsequent work in may customize the tubes that contribute to the orientation of the PVA chains for studying the issue you were curious about.

Figure R2. (a) SEM, (b) EDS, and (c) SAXS tests of the bionic hydrogel fiber.

3. Is there any difference in microstructure of the hydrogel fibers (PVA/PAA) when prepared in presence/absence of the metal ions and in presence/absence of salting out effect? SEM analysis may shed some light on this aspect.

Response: Thanks for the specific suggestion. SEM tests of hydrogel fibers (PVA/PAA) in the presence/absence of metal ions and in the presence/absence of salting out effect were supplemented and presented in Figure S1. With the step-by-step introduction of ionic coordination and crystalline domain crosslinking, the network structure of hydrogel fibers was gradually denser, implying the synergistic enhancement of molecular (chain) interactions by ionic coordination and crystalline domain crosslinking.

Figure S1 (updated). SEM tests of PA, PAZr, S-PA, and S-PAZr hydrogel fibers. PVA/PAA (PA) hydrogel fibers without ionic crosslinking and non-salting-out treatment, S-PVA/PAA (S-PA) hydrogel fibers with salting-out treatment, PVA/PAA/Zr⁴⁺ (PAZr) hydrogel fibers with ionic crosslinking, and S-PVA/PAA/Zr⁴⁺ (S-PAZr) hydrogel fibers with ionic crosslinking and salting-out treatment.

4. SAXS experiments analyze a small section of the sample. How homogeneous are the hydrogel fibers? What is the mean and standard deviation for the values provided for the distance between crystalline domains and the estimated radius of these domains?

Response: Good point. The homogeneity of the bionic hydrogel fibers was characterized by SEM and EDS tests. Due to the small pore size of the S-PAZr, its morphology could be seen only at ultra-high magnification. Therefore, we selected the center and edge regions of the hydrogel fibers for magnification comparison. As shown in Figure R3, the similar network structure and uniform elemental distribution implied good homogeneity of the bionic hydrogel fibers.

Figure R3. SEM and EDS tests of the S-PAZr for demonstrating homogeneity.

Further, the corresponding distance between crystalline domains (L) and the estimated radius of crystalline domains (R_g) were roughly obtained by multiple SAXS analyses as 7.39 ± 0.37 and 2.48 ± 0.16 , respectively.

The R_g could be calculated from the slope in the linear region of $\ln[I(q)]-q^2$ curve at low q values in the SAXS results according to Guinier equation (*Advanced Materials*. 2023, 35, 2301551; *Advanced Materials*. 2023, 35, 2209913):

$$I(q) = I(0)\exp\left(\frac{-q^2 R_g^2}{3}\right)$$

where $I(q)$ was the scattering intensity and $I(0)$ was the zero angle scattering intensity.

The distance (L) between adjacent crystalline domains was obtained from the Bragg equation (*Advanced Materials*. 2021, 33, 2102011; *Science Advances*. 2019, 5, eaau8528):

$$L = \frac{2\pi}{q_{max}}$$

where q_{max} was the critical vector corresponding to the peak intensity.

5. The SAXS scattering data have been compared at 0% and 300% strain. At least one more data set should be provided at higher strain% to confirm the trend.

Response: Thank you for the helpful comments. As requested by the reviewer, we have provided the SEM and higher strain (500% strain) SAXS data to reveal the toughening mechanism of the S-PAZr hydrogel fiber. As shown in Figure R4, the network structure of the S-PAZr would be oriented during stretching, which was also confirmed by the change of the 2D SAXS pattern from a circle to an ellipse. In the 1D SAXS curves, the scattering of the S-PAZr at 0.016 \AA^{-1} and 0.082 \AA^{-1} were mainly attributed to the interaction of PAA with cations and the semicrystalline lamellar domains of PVA, respectively (*Journal of Polymer Science Part B: Polymer Physics*. 2011, 49, 96-102; *Macromolecules*. 2017, 50, 6054-6063; *The Journal of Chemical Physics*. 2007, 127, 154908). When the S-PAZr was stretched from 0 to 500% strain, the scattering at 0.016 \AA^{-1} and 0.082 \AA^{-1} were shifted to 0.015 \AA^{-1} and 0.078 \AA^{-1} , respectively. It was implied that the carboxylic acid-cation interactions and PVA crystalline domains were damaged for dissipating energy during the stretching.

Figure R4. (a) SEM images of S-PAZr before and after stretching. (b) In situ stretching SAXS of S-PAZr.

6. There are numbering mistakes in Figure 4, which should be corrected.

Response: Thank you for the kind reminding. We have corrected Figure 4.

Figure 4 (updated). The BSE strategy was generalized to different combinations of Hofmeister effect-sensitive polymers and inorganic salts. (a) Hofmeister effect-sensitive polymers such as chitosan, agarose, and gelatin. (b-c) Mechanical properties of bionic hydrogel fibers prepared with 1 wt.% chitosan, 2 wt.% agarose, and 10 wt.% gelatin, respectively. (d) Stress-strain curves of bionic hydrogel fibers with different 0.3 M cationic crosslinking and saturated Na_2SO_4 -based ternary solvent treatment. (e) Stress-strain curves of bionic hydrogel fibers with 0.3 M Zr^{4+} crosslinking and different 1M sodium salts-based ternary solvent treatment. (f) Modulus and (g) toughness of bionic hydrogel fibers tuned by various ions. (h) According to the elastic modulus of bionic hydrogel fibers, the effect sequence of ions on bionic hydrogel fibers.

Reviewer #2 (Remarks to the Author):

In this study, inspired by the structure of spider silk, the authors constructed a biomimetic hydrogel fiber with high toughness and mechanical strength. The hydrogel is capable of adapting to low temperatures and dry environments, and is able to regenerate itself after water loss without loss of mechanical and electrochemical properties. The authors also made reasonable experiments to verify this conclusion. Therefore, the manuscript is recommended for publishing in Nature Communications after minor revisions as follows:

Response: We greatly appreciate the reviewer for the encouraging feedback on our manuscript and the valuable suggestions.

1.The introduction states that "it remains a challenge to achieve both high mechanical properties and environmental adaptability of hydrogel fibers.". However, the manuscript only demonstrates that the bionic hydrogel fibers have good performance at -40 °C and in dry environments. It is therefore recommended that more precise wording be used.

Response: We apologize for the imprecise wording. As suggested by the reviewers, we have revised the title of the manuscript to "Spider-silk-inspired, strong and tough hydrogel fiber with anti-freezing and water retention properties". In addition, the description of environmental adaptability in the manuscript and supplementary information has been revised accordingly with blue highlighting.

2.How did the authors determine the ratio of the inorganic salt/glycerol/H₂O ternary solvent? This doesn't seem to come across in the experimental section.

Response: Great point. An ideal inorganic salt/glycerol/H₂O ternary solvent should possess a high solubility of inorganic salt and a high percentage of glycerol, because a high percentage of salt and glycerol was beneficial for improving the mechanical properties, anti-freezing ability, and water retention capacity of hydrogel fibers (*Matter*. 2023, 6, 983-999; *Nature Materials*. 2020,

19, 1102–1109; *Materials Horizons*. 2021, 8, 351-369). Since inorganic salt was the main variable discussed in this manuscript, the priority order for the configuration ratio of inorganic salt/glycerol/H₂O ternary solvent was: inorganic salt > glycerol > H₂O. Based on Yue et al. reported that the kosmotropic inorganic salt (Na₂SO₄) had a high solubility in glycerol/water solution at a glycerol-to-water weight ratio of 2:8, and that continuing to increase the ratio of glycerol resulted in a significant decrease in the solubility of Na₂SO₄ (*The Chinese Journal of Process Engineering*. 1984, 4, 20-24). This conclusion was also confirmed by us. As shown in Figure R5, the Na₂SO₄/glycerol/H₂O ternary solvent with 2 mol/L Na₂SO₄ could be easily prepared by the binary solvent with a glycerol-to-water ratio of 2:8, whereas the binary solvent with a glycerol-to-water ratio of 3:7 was difficult to prepare the Na₂SO₄/glycerol/H₂O ternary solvent with 2 mol/L Na₂SO₄. Therefore, we selected a glycerol/H₂O binary solvent with a glycerol-to-water ratio of 2:8 to prepare the inorganic salt/glycerol/H₂O ternary solvent. In the experimental section, corresponding explanations were added to solve the possible confusion of readers (page 23, lines 418-421).

Figure R5. Solubility of Na₂SO₄ in solvents with glycerol-to-water weight ratios of 3:7 and 2:8.

3. It is suggested that on page 5, line 125, the words "Na₂SO₄/glycerol/H₂O" should be replaced by "Na₂SO₄/Gly/H₂O" for consistency with the text.

Response: We thank the reviewer for the kind reminder. Following the reviewer's suggestion, "Na₂SO₄/glycerol/H₂O" has been revised to "Na₂SO₄/Gly/H₂O" on page 5, line 108.

4. In Figure 4, the font of "Hofmeister effect" is wrong; on page 16, the format of " Δ " is not standardized; in Figure S6, the font of horizontal coordinates is not standardized. There are also some formatting errors, which the authors are also asked to double-check and revise.

Response: We apologize for some irregular fonts and formatting. We have corrected all of the above errors, e.g. Figure 4a and Figure S6. In addition, other formatting errors in the manuscript have been double-checked and revised. Corresponding revisions were highlighted in blue.

Figure 4 (updated). (a) Hofmeister effect-sensitive polymers such as chitosan, agarose, and gelatin.

Figure S8 (updated). Diameters of PAZr hydrogel after immersion in Gly/H₂O solutions of 1 M Na₂SO₄, NaCl, Na₃Cit, NaAc, Na₂HPO₄, and Na₂CO₃ for 24 h.

5. In the introduction section, it is recommended to provide a clearer and more comprehensive description of the background, purpose and significance of the study. It is also recommended to refer to some literature on hydrogel sensing and the Hofmeister effect (e.g., Adv. Mater. 2021, 33(11): 2007829; Adv. Funct. Mater. 2023, 33(12): 2213283; Nat. Commun. 2017, 8(1): 15911).

Response: We thank the reviewer for this constructive suggestion. Based on the reviewer's proposal, we have provided a clearer and more comprehensive description of the background, purpose and significance of the study, and the corresponding revisions have been marked in blue in the manuscript. Moreover, *Advanced Materials*. 2021, 33, 2007829 and *Advanced Functional Materials*. 2023, 33, 2213283 presented in-depth insights into the Hoffmeister effect, and *Nature Communications*. 2017, 8, 15911 exhibited an adaptive and freeze-resistant heterogeneous network organohydrogel, offering a new thought on improving the environmental tolerance of gels. These referenced works have been cited by us for improving our introduction.

6. On page 12 it says "The superior toughening effect of the BSE strategy requires the synergy of robust coordinated cations with kosmotropic salts...". How is this synergy achieved? Does it have any effect on the various properties of hydrogel fibers?

Response: We thank the reviewer for the valuable question. "The superior toughening effect of the BSE strategy requires the synergy of robust coordinated cations with kosmotropic salts...", this conclusion was speculated on the fact that only bionic hydrogel fibers treated with both high-valent cations and kosmotropic salts with less effect on ionic coordination, exhibited high mechanical properties (Figure 4d-g). The kosmotropic salts containing incompletely ionized anions caused an alkaline environment, which disrupted the ionic coordination of hydrogel fibers, resulting in a swelling that was unfavorable for the mechanical properties of hydrogel fibers (Figure S8). Obviously, the achievement of the synergistic effect required low mutual losses in the construction of ionic coordination or crystalline domain crosslinking to achieve the association of multilevel crosslinking structure for realizing the effect of $1+1 > 2$. Maybe the unclear description misled the reviewer, and we have revised this part for giving readers a clearer understanding (page 14, lines 274-277).

Figure 4 (updated). (d) Stress-strain curves of bionic hydrogel fibers with different 0.3 M cationic crosslinking and saturated Na₂SO₄-based ternary solvent treatment. (e) Stress-strain curves of bionic hydrogel fibers with 0.3 M Zr⁴⁺ crosslinking and different 1M sodium salts-based ternary solvent treatment. (f) Modulus and (g) toughness of bionic hydrogel fibers tuned by various ions.

Figure S8 (updated). Diameters of PAZr hydrogel after immersion in Gly/H₂O solutions of 1 M Na₂SO₄, NaCl, Na₃Cit, NaAc, Na₂HPO₄, and Na₂CO₃ for 24 h.

In addition, the synergy of ionic coordination and Hofmeister effect not only showed excellent effect in toughening hydrogel fibers, but also was significant in reducing the porosity of hydrogel fibers, as well as enhancing the energy dissipation of hydrogel fibers. The porosity of the hydrogel fibers was characterized by the SEM test in Figure S1. With the step-by-step introduction of ionic coordination and crystalline domain crosslinking, the network structures of the hydrogel fibers were progressively denser, suggesting a synergistic reduction of the porosity of the hydrogels by ionic coordination and crystalline domain crosslinking. As shown in Figure S4, the dissipated energy of the

PVA/PAA (PA) hydrogel fibers was 0.037 MJ/m^3 when performing a 100% strain loading-unloading cycle. When ionic coordination and crystalline domain crosslinking were introduced, the dissipated energy of hydrogel fibers increased to 1.53 MJ/m^3 and 0.67 MJ/m^3 , respectively. Notably, the dissipated energy of the S-PAZr hydrogel fiber with both ionic coordination and crystalline domain crosslinking were 10.06 MJ/m^3 , implying the synergistic enhancement of dissipated energy by ionic coordination and crystalline domain crosslinking.

Figure S1 (updated). SEM tests of PA, PAZr, S-PA, and S-PAZr hydrogel fibers. PVA/PAA (PA) hydrogel fibers without ionic crosslinking and non-salting-out treatment, S-PVA/PAA (S-PA) hydrogel fibers with salting-out treatment, PVA/PAA/ Zr^{4+} (PAZr) hydrogel fibers with ionic crosslinking, and S-PVA/PAA/ Zr^{4+} (S-PAZr) hydrogel fibers with ionic crosslinking and salting-out treatment.

Figure S4 (updated). (a-b) Loading-Unloading curves and (c) dissipation energy of PA, PAZr, S-PA, and S-PAZr hydrogel fibers.

7. What is the lifetime of this hydrogel fiber? Is there a limit to the number of self-regenerations? Do other external environmental factors affect the performance of the material? How did the authors avoid these problems?

Response: We thank the reviewer for the insightful question. For the lifetime of this hydrogel fiber, we evaluated it by the change of mechanical and electrical properties of the hydrogel fiber after long-term storage at 90% RH. As shown in

Figure R6a-b, the mechanical and electrical properties of hydrogel fibers after 7 and 30 days of storage at 90% RH were almost similar to the original sample, implying the hydrogel fibers could be stored for a long-term period of time under suitable storage conditions. In addition, we performed more self-regeneration tests on the hydrogel fibers to investigate whether there were limits to the number of self-regenerations of hydrogel fibers (Figure R6c). Before three self-regeneration tests, the hydrogel fibers maintained favorable mechanical and electrical properties. As the number of self-regeneration times increased, the mechanical properties of the hydrogel fibers gradually fluctuated, whereas the conductivity remained nearly stable, which may be attributed to small defects detrimental to the mechanical properties of the hydrogel fibers caused by multiple dehydration processes. Nevertheless, hydrogel fibers kept good mechanical and electrical properties during the first three self-regenerations. Corresponding revisions were marked in blue to the manuscript and supplementary information.

Other external environmental factors such as light, high temperature, vacuum, etc. may affect PVA crystallinity and water content, leading to changes in performances of hydrogel fibers. We mainly avoided the influence of these external environments on hydrogel fibers by preparing and testing them on the spot and applying vacuum silicone grease. Of course, resistance to these external environments could also be achieved by designing the gel fibers with temperature-insensitive polymers and volatility-resistant solvents (e.g., glycerol, deep eutectic solvents, ionic liquids, etc.) (*Nature Materials*. 2020, 19, 1102–1109; *Nature Communications*. 2022, 13, 6671; *Advanced Functional Materials*. 2022, 32, 2203988).

Figure R6. (a) Mechanical properties and (b) conductivity of the bionic hydrogel fiber for long-term storage at 90% RH. (c) Electrical and mechanical properties of self-regenerated bionic hydrogel fibers.

8. It is recommended that the format of references be standardized, e.g., some authors are fully marked out and some are omitted. It is recommended that the full text be revised in accordance with the journal format.

Response: Thank you for the kind reminding. some authors were fully marked out and some were omitted, which was attributed to the journal's requirement to list only the first author when more than six authors were included in the reference. Fortunately, we did find some errors in the references after being reminded by the reviewer. Based on the reviewers' comments, we have corrected the references and revised the full text according to the journal format.

Reviewer #3 (Remarks to the Author):

In the manuscript entitled “A Strong and Tough Hydrogel Fiber with Environmental Adaptability Inspired by Spider Silks”, authors constructed a hydrogel fiber with elaborated ionic crosslinking and crystalline domains inspired by the multilevel adjustment of spider silk network structure with ions. In addition, the introduction of inorganic salt/glycerol/H₂O ternary solvent allowed these hydrogel fibers to adapt to low-temperature and dry environments with almost no loss of mechanical and electrical properties. Furthermore, this work provided ideas to fabricate hydrogel fibers for flexible sensing applications. However, there are still many problems to be solved. Therefore, the referee didn't recommend publishing it in Nature Communications.

Response: We appreciate the reviewer for their carefully reading and providing insightful feedback to help us further improve our manuscript. We thank the reviewers for recognizing the design thinking and importance of our work. We have answered all your concerns and technical questions below, and we believe the quality and scientific content of our work has significantly improved. We hope that our revisions could address the reviewer's concerns and convince the reviewer to support our manuscript for publication in Nature Communications.

Before responding to the reviewer's comments on improvement and correction, we will try to demonstrate why our work is of high enough novelty and impact, as well as being interesting for the broad readership of Nature Communications.

First, we are extremely grateful that editor has allowed us the chance to highlight the novelty and innovation of the work, as well as to reviewer #1 and reviewer #2 for highly recognizing the innovation and systematization of our work. Unlike most of the spider-silk-inspired materials that focus only on the β -crystalline domain, we concentrate attention on the important factor regulating the spinning process of spider silk, i.e., a series of cations and anions, which

could adjust the multilevel network structure and various performances of spider silk through ionic coordination and Hofmeister effects, demonstrating a promising salt-regulated structure-performance paradigm (*Nature Communications*. 2023, 14, 1370; *Nature Communications*. 2019, 10, 5293; *Advanced Science*. 2022, 9, 2103965; *Soft Matter*. 2006, 2, 448-451). We specifically highlighted the salt-regulated structure-performance paradigm from spider silks and migrated it to the design of hydrogel fibers. To the best of our knowledge, our study was the first attempt to construct high-performance hydrogel fibers by utilizing the salt-regulated structure-performance paradigm of spider silks, which has never been demonstrated before. Based on the bionic structure engineering (BSE) strategy developed from the salt-regulated structure-performance paradigm, we overcame the poor mechanical properties and environmental instability of hydrogel fibers due to their inherent weak molecular (chain) interactions, ensuring real-world applications of hydrogel fibers. We believe that the BSE strategy based on the salt-regulated structure-performance paradigm could provide new ideas for the development of bionic materials.

In addition, bionic hydrogel fibers developed by utilizing the BSE strategy exhibited excellent and widely tunable mechanical properties. The bionic hydrogel fibers showed superior mechanical properties comparable to spider silk (Figure R7a). For example, stress levels of 24.43 ± 2.11 MPa, ultimate strains of $1155.74 \pm 165.42\%$, and toughness of 162.25 ± 21.99 MJ/m³. Notably, bionic hydrogel fibers allowed easily customize of their mechanical properties by regulating the ionic crosslinking and crystalline domains, especially the elastic modulus could be raised from gel to plastic levels ($0.27 \pm 0.025 \sim 118.53 \pm 5.49$ MPa). Clearly, the mechanical properties of bionic hydrogel fibers were not only superior to other currently reported hydrogel fibers, but its customizable mechanical properties were convenient and essential for the application of hydrogel fibers (Figure R7b-c). Furthermore, as shown in our manuscript, the BSE strategy based on the salt-regulated structure-performance paradigm

could be generalized to other combinations of polymers and inorganic salts, which allowed it to be applied to more systems to promote their applications. Additionally, it should be explained that since the most prominent characteristic of natural spider silk was its mechanical properties. Therefore, our discussion focused on the mechanical properties of bionic hydrogel fibers and other additional properties such as anti-freeze, water retention and self-regeneration were weakly discussed.

Overall, we would like to emphasize that the salt-regulated structure-performance paradigm from spider silks was a significant and major departure from previous spider-silk-inspired strategies that focused only on the β -crystalline domain. Moreover, this was the very first highlighted the salt-regulated structure-performance paradigm of spider silks and utilized to improve the mechanical properties and environmental tolerance of hydrogel fibers, to the best of our knowledge. We apologized for not emphasizing and clarifying the novelty enough in the previous manuscript, and we have correspondingly revised the manuscript to hopefully avoid any confusion. Please feel free to let us know if you have any other concerns.

Figure R7. (a) Ashby plots of toughness versus tensile strain of S-PAZr hydrogel fibers and other high-toughness materials. Ashby plots of (b) toughness versus tensile strain, (c) toughness versus elastic modulus of S-PAZr hydrogel fibers and other reported hydrogel fibers.

1. Despite the strong mechanical properties of the hydrogel fibers, the sensing performance is not outstanding when it is used as sensing application.

Response: Great point. We thank the reviewers for recognizing the mechanical properties of our hydrogel fibers. It should be emphasized that the main goal of our work was to overcome the poor mechanical properties and environmental instability of hydrogel fibers due to their weak intrinsic molecular (chain) interactions to promote the real-world applications of hydrogel fibers. The sensing application of bionic hydrogel fibers was only to show its usability.

On the other hand, we agree with the reviewer that the sensitivity (0.88) of our hydrogel fiber-based strain sensor was relatively low, but it was superior to currently reported hydrogel fiber-based strain sensors with a monitoring range of up to 800% (Figure R8). For the sensing performance of hydrogel fiber. According to our knowledge, the intrinsic sensing performance of ionic flexible sensors was mainly dependent on the material shape (*Nature Reviews Materials*. 2018, 3, 125–142). Therefore, it was normal for fiber-shaped flexible sensors to exhibit low sensitivity (gauge factor < 2). Although fiber-shaped sensors with undesirable sensitivity, the weavability of the fibers allowed them to show great potential in the flexible sensing. Based on the weavability of fibers, they could not only be used to construct multi-dimensional flexible sensors with multiple functions from the bottom up, but also to adjust the sensitivity of flexible sensors by changing the weaving method (*Advanced Science*. 2023, 10, 2305226; *Chemical Engineering Journal*. 2023, 455, 140796; *Advanced Materials*. 2020, 32, 2003897). In addition, common methods for improving the sensitivity of hydrogel bulk-based sensors such as constructing multilayer structures, doping electrical conductors, and designing surface microstructures were also applicable for hydrogel fiber-based sensors, which nicely addressed the concerns about the low sensitivity of hydrogel fiber-based sensors (*Advanced Fiber Materials*. 2023, 5, 1643–1656; *Materials Horizons*. 2023, 10, 3569-3581; *Journal of Materials Chemistry A*. 2021, 9, 12265-12275; *Materials Horizons*. 2021, 8, 2088-2096).

Figure R8. Comparison of sensing performance of hydrogel fiber flexible sensors. PVA/SA/Ca²⁺ (*Small*. 2024, 20, 2305951), HPH@PAPAM (*Chemical Engineering Journal*. 2023, 458, 141393), SA/Ca²⁺/P(HEA-co-PEGDA) (*Advanced Materials*. 2020, 32, 1906994), PMON (*Journal of Materials Chemistry A*. 2021, 9, 12265-12275), HPIF/Ca²⁺ (*ACS Applied Materials & Interfaces*. 2021, 13, 43323–43332), PNA (*Nano Energy*. 2020, 78, 105389), and S-PAZr (this work).

2. The authors should be strict about some terms, such as the description of relative intensity in XPS.

Response: Thank you for the kind reminding. According to the reviewer's suggestion, we have checked and revised the terminology in the full text several times by reading through specialized books and references, and the corresponding revisions were marked in blue. For example, FTIR and XPS analyses. As shown by FTIR spectra, the -OH characteristic peak of the PA hydrogel fiber gradually shifted from 3369 cm⁻¹ to 3322 cm⁻¹ with the introduction of Zr⁴⁺ and salting-out treatment, implying their synergistic enhanced the hydrogen bonding interactions in the S-PAZr hydrogel fiber (Figure 2a). In addition, the characteristic peaks of -COO⁻ at 1575 cm⁻¹ and the characteristic peaks in 850-950 cm⁻¹ could be attributed to the interaction of -COO⁻ with ions and the introduction of the ternary solvent, which was reconfirmed by high-resolution O 1s XPS spectra (Figure 2b) (*Chemistry of Materials*. 2022, 34, 1392-1402; *Advanced Functional Materials*. 2020, 30, 1910387). Notably, the number of Zr-O bonds in the PAZr hydrogel fiber

decreased from 10.31% to 5.53% after salting-out treatment, while the binding energy exhibited an unremarkable shift in high-resolution Zr 3d XPS spectra, indicating that the number of Zr⁴⁺ crosslinking in the S-PAZr hydrogel fiber was reduced after the salting-out treatment, while the Zr⁴⁺ coordination state was not significantly changed (Figure 2c) (*Advanced Functional Materials*. 2023, 33, 2307402).

Figure 2 (updated). (a) FTIR spectra of PA, PAZr, S-PA, and S-PAZr hydrogel fibers. (b) O 1s XPS spectra of PA, PAZr, S-PA, and S-PAZr hydrogel fibers. (c) Zr 3d XPS spectra of PAZr and S-PAZr hydrogel fibers.

3. Compared with other works (*Nature Communications*, 2022, 13:3408; *ACS Appl. Mater. Interfaces*, 2020, 12, 25353–25362), the hydrogel fibers have low conductivity at low temperature.

Response: We thank the reviewer for the insightful question. We agree with the reviewer that the conductivity of the bionic hydrogel fiber was low. However, we would like to clarify that the focus of the discussion in our manuscript was not on conductivity, but rather on the spider-silk-inspired salt-regulated structure-performance paradigm for the enhancement of mechanical properties and environmental tolerance of the hydrogel fiber. Conductivity was only one of the indexes used to evaluate environmental tolerance.

Moreover, high mechanical properties inevitably led to low ionic conductivity, which was a classical trade-off, and we were also trying to overcome this challenge in the subsequent work. The reviewer-recommended references (*Nature Communications*. 2022, 13, 3408; *ACS Applied Materials & Interfaces*. 2020, 12, 25353–25362) provided means for improving the

mechanical properties and conductivity of the hydrogel by using supramolecular engineering strategies and doping ionic liquid, respectively, which provided some thoughts for us and were cited in the relevant parts of the manuscript.

4. For the part of structural evolution and toughness mechanism, more characterization is needed to verify its accuracy, such as SEM, TEM and more.

Response: We thank the reviewer for the helpful comments. According to the reviewers' comments, we have supplemented the SEM of PVA/PAA (PA) hydrogel fibers without ionic crosslinking and non-salting-out treatment, S-PVA/PAA (S-PA) hydrogel fibers with salting-out treatment, PVA/PAA/Zr⁴⁺ (PAZr) hydrogel fibers with ionic crosslinking, and S-PVA/PAA/Zr⁴⁺ (S-PAZr) hydrogel fibers with ionic crosslinking and salting-out treatment. In addition, SEM and SAXS analyses of S-PAZr hydrogel fibers during stretching were performed to further verify the structural evolution and toughening mechanism of bionic hydrogel fibers.

As shown in Figure R9a, with the step-by-step introduction of ionic coordination and crystalline domain crosslinking, the network structure of hydrogel fibers was gradually denser, implying the synergistic enhancement of molecular (chain) interactions by ionic coordination and crystalline domain crosslinking. Further, SEM and SAXS tests of S-PAZr hydrogel fibers during stretching were presented in Figure R9b-c. The network structure of S-PAZr would be oriented during stretching, which was also confirmed by the change of the 2D SAXS pattern from a circle to an ellipse. In the 1D SAXS curves, the scattering of the S-PAZr at 0.016 Å⁻¹ and 0.082 Å⁻¹ were mainly attributed to the interaction of PAA with cations and the semicrystalline lamellar domains of PVA, respectively (*Journal of Polymer Science Part B: Polymer Physics*. 2011, 49, 96-102; *Macromolecules*. 2017, 50, 6054-6063; *The Journal of Chemical Physics*. 2007, 127, 154908). With the gradual increase in tensile strain, the scattering at 0.016 Å⁻¹ and 0.082 Å⁻¹ were shifted to 0.015 Å⁻¹ and 0.078 Å⁻¹,

respectively. It was implied that the carboxylic acid-cation interactions and PVA crystalline domains were damaged for dissipating energy during the stretching. during the stretching. We believe that the above characterizations provided more detailed supplements to the structural evolution and toughening mechanism of the bionic hydrogel fibers, which were helpful to strengthen the science and systematization of our work.

Figure R9. (a) SEM images of PA, PAZr, S-PA, and S-PAZr hydrogel fibers. (b) SEM images of S-PAZr before and after stretching. (c) In situ stretching SAXS of the S-PAZr hydrogel fibers.

REVIEWERS' COMMENTS

Reviewer #1 (Remarks to the Author):

In my opinion, the authors have sufficiently addressed the concerns raised in this revised version of the manuscript. The manuscript can be accepted as it is.

Reviewer #2 (Remarks to the Author):

Initial comments have been responded to and resolved.

Reviewer #3 (Remarks to the Author):

In the manuscript entitled "A Strong and Tough Hydrogel Fiber with Environmental Adaptability Inspired by Spider Silks", authors constructed a hydrogel fiber with elaborated ionic crosslinking and crystalline domains inspired by the multilevel adjustment of spider silk network structure with ions. In addition, the introduction of inorganic salt/glycerol/H₂O ternary solvent allowed these hydrogel fibers to adapt to low-temperature and dry environments with almost no loss of mechanical and electrical properties. Furthermore, this work provided ideas to fabricate hydrogel fibers for flexible sensing applications. Therefore, the referee recommend publishing it in Nature Communications.